# Modulation of Immunity, Antioxidant Status, Performance, Blood Hematology, and Intestinal Histomorphometry in Response to Dietary Inclusion of *Origanum majorana* in Domestic Pigeons’ Diet

**DOI:** 10.3390/life13030664

**Published:** 2023-02-28

**Authors:** Hala Y. Amer, Rasha I. M. Hassan, Fatma El-Zahraa A. Mustafa, Ramadan D. EL-Shoukary, Ibrahim F. Rehan, František Zigo, Zuzana Lacková, Walaa M. S. Gomaa

**Affiliations:** 1Department of Animal Nutrition and Clinical Nutrition, Faculty of Veterinary Medicine, New Valley University, El-Kharga 72511, Egypt; 2Department of Nutrition and Clinical Nutrition, Faculty of Veterinary Medicine, Assiut University, Assiut 71515, Egypt; 3Department of Cell and Tissues, Faculty of Veterinary Medicine, Assiut University, Assiut 71515, Egypt; 4Department of Animal Hygiene, Faculty of Veterinary Medicine, New Valley University, El-Kharga 72511, Egypt; 5Department of Husbandry and Development of Animal Wealth, Faculty of Veterinary Medicine, Menoufia University, Shebin Alkom, Menoufia 32511, Egypt; 6Department of Pathobiochemistry, Faculty of Pharmacy, Meijo University, Yagotoyama 150, Tempaku-ku, Nagoya-shi 468-8503, Japan; 7Department of Nutrition and Animal Husbandry, University of Veterinary Medicine and Pharmacy, Komenského 73, 04181 Košice, Slovakia

**Keywords:** *Origanum majorana* powder, performance, antioxidant, immune response, hematology

## Abstract

This experiment was conducted to evaluate the effect of adding *Origanum majorana* (OM) powder to domestic pigeon diets on growth performance, feeding and drinking behaviour, blood hematology, blood biochemical parameters, blood inflammatory and oxidative markers, carcass characteristics, the weights of lymphoid organs, and and intestinal cecal, and bursa of Fabricius histology. A random distribution of fifty-four unsexed pigeon squabs (30 days old, average body weight; 321 g ± 7.5) into three groups was done. The first group was fed the grower basal diet without adding OM powder, while OM powder was added at levels of 0.5 and 1% to the basal diets of the second and third groups, respectively. The changes in growth performance parameters and feeding and drinking behavior under OM powder’s effect were insignificant. However, the lymphoid organs (spleen and thymus) significantly increased in weight (*p* < 0.05) in the OM-fed groups. Moreover, blood examination showed positive responses to OM powder in terms of blood cell counts (RBCs andWBCs), and the values of hemoglobin, hematocrit, mean corpuscular volume, lymphocyte numbers, levels of globulin, and glutathione peroxidase enzyme were significantly increased. The numbers of heterophils, the ratio of heterophil to lymphocyte, malondialdehyde levels were reduced (*p* < 0.05). Histomorphometry examination revealed increases in intestinal villi height, cecal thickness, and bursal follicle area and number. These results indicated that adding OM powder to the pigeon diet may improve their immunity, increase their antioxidant status, and correct some hematological disorders.

## 1. Introduction

Recently, the animal industry has expanded worldwide, including the production of livestock, companion animals, and poultry. Among these sectors, the most significant contributor to this expansion was poultry production [1]. It is known that poultry production is characterized by its lower cost, better feed conversion, and fewer associated environmental and health problems than other livestock productions [2,3]. Because of all these factors contributing to the growing population, poultry production has grown with rapid conversion to the commercial production type in developing countries [4]. Therefore, for improving the performance, immune response, and health of poultry, it was essential for researchers to examine new feed additives and include them into poultry diets. Herbal plants and their extracts are considered promising additives in poultry production; they can be used as growth promoters and immune modulators as replacements for antibiotics, which have adverse effects on poultry [5,6].

*Origanum majorana* [OM], or sweet marjoram, is a creeping aromatic medicinal herbal plant [7] that belongs to the family Lamiaceae. It is very popular in Western Asia and North Africa [8]. Because of its richness in phenolic compounds, flavonoids, and essential oils, with borneol, terpinene, pinene, sabinene, and terpineol contents [9], *Origanum majorana* is characterized by its antioxidant, antibacterial, antifungal, antiseptic, analgesic, immune modulator, and metabolism-inducing properties [10,11]. Moreover, OM extract could protect against renal and liver damage [8], lead acetate injury [12], and hyperlipidemia [13].

The broiler response to diets supplemented with prebiotics, probiotics, or herbal mixtures (*Origanum majorana*, *Carum carvi*, and *Foeniculum vulgare*) as alternatives to antibiotics indicated that the herbal mixture group recorded the highest productive performance [14]. Moreover, Saleh et al. [15] added a mixture of OM and another medicinal herbal plant to laying hen diets and noticed an improvement in their productivity and performance, including the feed conversion ratio and egg quality and quantity.

Because of its highly palatable and delicious meat (which indicates high nutritional value), effortless management and rearing, and rapid weight gain, the marketing of domestic pigeons is very common in Egypt [16]. To our knowledge, few studies have been conducted in order to study the nutritional, behavioral, antioxidant, and immunomodulatory impacts of adding OM powder to the pigeon diet.

In this study, we hypothesized that adding OM powder to pigeon diets may modulate their growth performance, feeding and drinking behavior, immune response, antioxidant status, and intestinal absorption in a desirable manner. Therefore, this experiment studied the effect of adding OM to the pigeon diet on performance, feeding and drinking behavior, carcass parameters and lymphoid organ weights, blood hematology, blood biochemical parameters, antioxidant and inflammatory markers, and intestinal, cecal, and bursal histomorphometry.

## 2. Materials and Methods

### 2.1. Origanum majorana Powder

*Origanum majorana* (OM) powder was purchased from a commercial source (Organic, Natural Oil Factory, Assiut, Egypt), and prepared and analyzed (using the methods described by the AOAC [17]) in the Animal Nutrition and Clinical Nutrition Lab., New Valley University, Egypt. The OM chemical analysis indicated that it contains 95.5% dry matter (DM), 3.3% ether extract (EE), 14% crude protein (CP), 10.3% ash, and 17.5% crude fiber (CF) using the following official methods: AOAC 930.15, AOAC 920.39, AOAC 984.13, AOAC 942.05, and AOAC 978.10. Metabolizable energy (ME) (2712 Kcal/Kg diet) was calculated based on the chemical composition, as described by the NRC [18]. In addition, the active principles of OM powder were analyzed in the Chemistry Lab., Faculty of Science, Assiut University, Assiut (see Appendix A). The active components included thymol (4.2%), terpineol contents (alpha-terpineol 3.6%, alpha-terpinene 6.8%, alpha-terpinolene 1.5%, and gamma-terpinene 5.5%), carene (1.1%), caryophyllene (1.5%), alpha-phellandrene (2.07%), aminopropyl phenol (0.09%), fluoro-5-ethyl phenol (0.2%), (1-Pyrrolyl) phenol (1.7%), 5-methyl phenol (0.14%), and cathine (0.02%). The previous studies [19,20] were used to decide the inclusion level of OM powder in the diets.

### 2.2. Birds, Diets, and Design

Fifty-four unsexed pigeon squabs (age: 30 days; average body weight; 321 g ± 7.5) from a local source (El-Matieuh rural villages—Assiut, Egypt) were randomly distributed into three groups (n = 18, 3 replicates, n = 6/group). The grower basal diet without supplementing OM powder was offered for the 1st group, while OM was added to the basal diets of the 2nd and 3rd groups at levels of 5 and 10 g/kg diet, respectively. The mashed form of the diet was used. The ingredients of the grower basal diet, which was formulated based on the recommendations of [21,22], are shown in Table 1. The temperature was adjusted according to the bird’s needs (18–23 °C). Natural and mechanical ventilations were supplied. Free access to both water and feed was provided. The schematic cartoon (Figure 1) of the experimental study was designed by BioRender.com.

### 2.3. Growth Performance

The body weight was recorded for each bird at the study beginning. After that, body weight (individual and cumulative), along with the feed intake of pigeon squabs, was recorded weekly. The feed conversion ratio (FCR), relative growth rate (RGR), and European production efficiency index (EPI) were calculated [23,24].

### 2.4. Feeding and Drinking Behavior Assessment

During the experiment, feeding and drinking behaviors were observed [25]. Pigeons involved in eating behavior (act\30 min) were recorded by observing their contact with feed and water, following the recommendation of Spudeit et al. [26].

### 2.5. Carcass Parameters and Lymphoid Organs

The experimental period was 45 days. At the experimental end, 3 birds per group were euthanized by slaughter after their random selection and weighing. The lymphoid organs (spleen, bursa of Fabricius, and thymus), liver, heart, and gizzard were weighed and expressed as a percentage of the live body weight [27].

### 2.6. Blood Examination

During slaughtering, blood was collected from the cervical vein and preserved in heparinized and non-heparinized tubes (Vacutainer, Becton Dickinson, Stuart, FL, USA).

### 2.7. Blood Hematology

The heparinized tubes were used for evaluating red blood cells (RBCs), white blood cells (WBCs), blood hemoglobin (Hb), mean corpuscular volume (MCV), mean corpuscular hemoglobin concentration (MCH), hematocrit value (HCT), and differential white blood cell count. The ratio of heterophils/lymphocytes was calculated. Using a hemocytometer and staining blood films with the Wright–Giemsa stain, numbers of RBCs and WBCs were counted.

### 2.8. Blood Biochemical Parameters

The blood samples in the other tubes were centrifuged for 15 min, at 3000 rpm at 4 °C, and kept at −20 °C till further analysis. Total proteins, albumin, globulin, total cholesterol, urea, and creatinine were determined by using commercial kits (Biotechnology Company, Assiut, Egypt).

### 2.9. Serum Inflammatory and Oxidative Markers

For inflammation detection, tumor necrosis factor α (TNF-α) and interleukin 6 (IL6) were determined by an ELISA Kit for chicken (Biotechnology Company, Assiut, Egypt). The malondialdehyde (MDA) and glutathione peroxidase (GPx), as oxidative markers, were measured by commercial colorimetric kits) Biotechnology Company, Assiut, Egypt) using a spectrophotometer (Unico UV 2000; Spectra Lab Scientific Inc., Alexandria, VA, USA).

### 2.10. Histomorphometry Analyses

Eight pigeons were randomly chosen to collect samples from the intestine (duodenum), cecum, and bursa of Fabricius. Immediately after slaughtering, samples were dissected, fixed in Bouin’s fluid, alcohol-dehydrated, cleared in methyl benzoate, and paraffin wax-embedded. After that, cutting at 4–5 μm thickness and staining with Harris hematoxylin were done [28]. Measurements of duodenal villi height/um, wall thickness/um, cecal muscle thickness/um, and follicle numbers/500 um and follicle area/um^2^ of the bursa of Fabricius were done using ImageJ software. Measurement data are described as means ± SDM.

### 2.11. Statistical Analyses

*Origanum majorana’s* effects on performance, behavior, carcass characteristics, blood hematology, blood parameters, and inflammatory and oxidative markers in pigeons were analyzed using SPSS [26.0]. For treatment comparison, Duncan’s multiple range test was used. The 5% level was used as an indication of significance [29]. The statistical model was Y_ij_ = μ + T_i_ + E_ij_, where Y_ij_ = response variables; μ = the overall mean; T_i_ = treatment effect; E_ij_ = the experimental error.

## 3. Results

### 3.1. Growth Performance Parameters

*Origanum majorana* powder’s effect on the performance of pigeon squabs is shown in Table 2. A numerical increase in growth performance parameters under the OM powder effect was observed. However, no significant differences were detected among groups in terms of body weight, weight gain, feed intake, feed conversion, production index, or relative growth rate.

### 3.2. Feeding and Drinking Behavior

The assessment of pigeons’ feeding and drinking behavior under the effect of *Origanum majorana* powder is indicated in Figure 2. Adding OM powder increased the birds’ feeding and drinking acts, but this increase was insignificant.

### 3.3. Carcass Parameters and Lymphoid Organs

The effect of *Origanum majorana* powder on the carcass parameters and lymphoid organs is presented in Table 3. The dressing percentages for liver, gizzard, bursa, and heart did not show any significant changes among the experimental groups. However, there was a significant increase in spleen and thymus relative weights (*p* ≤ 0.05) in the OM powder groups.

### 3.4. Blood Examination

#### 3.4.1. Hematological Parameters

The *Origanum majorana* powder effect on blood hematology is shown in Table 4. Red blood cells [RBCs], hemoglobin [Hb], hematocrit [HCT], mean corpuscular volume [MCV], white blood cells [WBCs], and lymphocyte % showed higher values with OM powder [*p* < 0.05]. While heterophils % and heterophils to lymphocyte ratio were significantly [*p* < 0.01] decreased.

#### 3.4.2. Blood Biochemical Parameters, Serum Inflammatory Markers, and Oxidative Markers

Blood parameters and markers affected by adding *Origanum majorana* powder to pigeon diets are presented in Table 5. Supplementation of OM powder had no effect on serum total protein, albumin, creatine, or urea levels. In contrast, globulin was higher [*p* < 0.05], and the albumin-to-globulin ratio was lower [*p* < 0.05] in the treated groups. Moreover, adding OM powder to pigeon diets at both levels reduced [*p* < 0.05] the cholesterol level.

Interleukin 6 was not affected, while tumor necrosis factor tended to be [*p* = 0.09] increased by adding OM powder. In addition, serum oxidative markers, malondialdehyde, and glutathione peroxidase enzyme were significantly [*p* < 0.01] decreased and increased, respectively.

### 3.5. Histomorphometry Examination

#### 3.5.1. Duodenal and Cecal Histomorphometry

A slight increase in intestinal villi length [about 510.134 ± 6.239 μm] was observed in the group fed 1% OM powder. However, intestinal villi height was nearly similar in the control group [about 504.036 ± 31.292 μm], and the group provided 0.5% OM powder [about 504.713 ± 4.021 μm] [Figure 3A–C and Figure 4].

The cecal wall thickness was increased with both levels of OM [about 1699.357 ± 4.468 μm and 1426.958 ± 8.336 μm with 0.5 and 1% OM, respectively] in comparison with the control [about 1321.432 ± 7.518 μm]. In addition, a slight increase in muscular layer thickness was detected in the 0.5% OM group [nearly 109.639 ± 1.426 μm] and 1% OM group [nearly 110.050 ± 6.347 μm] in comparison with the control group [nearly 107.265 ± 3.050 μm] [Figure 3D–F and Figure 4].

#### 3.5.2. Bursal Follicle Histomorphometry

The number of follicles was about 8.8 follicles per 500 μm in the control group. By adding OM powder, the number was increased to about 9 follicles per 500 μm with 0.5% OM and about 12 follicles per 500 μm with 1% OM. Moreover, the area of the follicle in the control group was about 249.500 ± 3.190 μm^2^ and increased to about 251 ± 5.332 μm^2^ in the 0.5% OM group. The largest follicle area was demonstrated with a 1% OM group [about 254.500 ± 5.816 μm^2^] [Figure 4 and Figure 5].

## 4. Discussion

### 4.1. Effect of Origanum majorana Powder on Growth Performance Characteristics

The numerical values showed that pigeons fed OM powder had higher feed intake, body weight, weight gain, and relative growth rate than the control group; no significant differences were detected among the three groups. Our results agreed with those obtained by Khattab et al. [30]; they investigated feeding different levels of *Origanum majorana*, *Pimpinella anisum*, and *Mentha piperita* in relation to growth performance improvements in broiler chicks. They indicated that neither feed conversion nor body weight were affected by adding *Origanum majorana* to the broiler diet. Moreover, Ali [19] reported a reduction in the daily feed intake of broilers by supplementing *Origanum majorana* at levels of 0.5, 1.0, and 1.5%. Contrary to our results, Shawky et al. [20] and Abdel-Wahab [31] indicated that adding dietary *Origanum majorana* to broiler diets improved their weight and weight gain.

No clear explanation was found for the variant effect of *Origanum majorana* on growth performance among the different research works. Still, it may be related to the variation in the level used in each experiment or other factors, such as stress. Vase-Khavari et al. [32] indicated that the efficacy of herbal plants and probiotics is correlated to different factors, such as their level and concentration used, the composition of the diet, environmental factors, and the hygiene of the poultry houses.

### 4.2. Effect of Origanum majorana Powder on Feeding and Drinking Behavior

Bird physiological conditions, diet composition, feeder space, and heat stress can affect birds’ feeding and drinking behavior [33,34,35]. The assessment results for feeding and drinking behavior were consistent with the performance results, as OM powder increased feeding and drinking behavior, but this increase was insignificant. Ramadan [36] and Harrington et al. [37] indicated that aromatic herbs and their extracts could increase feeding behavior in poultry. Ramadan [36] reported the presence of a negative correlation between fear and feeding behaviors in poultry, as a decrease in the fear response will increase feeding behavior. Scientists suggested that aromatic plants have a depressing effect on neural activity through the activation of GABA receptors, resulting in reduced fear behavior and increased feeding times [36,37,38]. The fear response was not assessed in our experiment.

### 4.3. Effect of Origanum majorana Powder on Carcass Characteristics and Lymphoid Organs

The dressing percentage and liver, heart, gizzard, and the bursa of Fabricius did not differ among pigeons supplemented with OM powder and the control group. The absence of the OM effect on carcass traits was expected, because of the similar growth performances among the three groups. Several studies reported that adding herbal plants either in powder or oil extract did not affect broilers’ internal organ weights [39]. Moreover, Shawky et al. [20] reported that the weights of the liver, heart, and gizzard did not show any significant difference when dietary supplementation of OM was used in broiler diets.

The avian immune response can be affected by several extrinsic or intrinsic factors; one of the significant extrinsic factors affecting bird immunity is the diet and its composition [40]. The lymphoid organs responsible for avian immunity include primary and secondary organs. The primary organs are the thymus and bursa of Fabricius [41]. These organs are the sites for maturation, differentiation, and immunocompetence of T and B types of lymphocytes [42]. Functional T and B cells depart from the primary to the secondary lymphoid organs, including the bone marrow and spleen [43]. Ahsan et al. [44] indicated that the relative weight of lymphoid organs reflects the bird’s immunity status. In our experiment, the thymus and spleen showed significant increases in their weights with the OM powder supply, which means that the pigeon immune responses were improved under the effect of *Origanum majorana*. The impact of OM may be related to the presence of flavonoids and phenolics, with their antibacterial, antioxidant, and immune-modulating effects. However, Ali [19] indicated that the spleen weight in broilers was not affected by adding variant *Origanum majorana* powder levels.

### 4.4. Effect of Origanum majorana Powder on Blood Hematology

Nutrition’s effect on bird physiology and metabolism can be indicated by examining blood hematology and biochemical blood parameters [45]. As a result, adding *Origanum majorana* to birds’ diet affected the pigeons’ hematology. *Origanum majorana* powder significantly increased both RBC (with 1% OM) and WBC counts, which are responsible for oxygen transfer and protection against infection, respectively. The elevation in the numbers of RBCs with OM may refer to its antioxidant activity, preventing lipid peroxidation in blood cell membranes. Moreover, the increased WBC number may be related to thymol (active component in OM), which is responsible for immune response enhancement [46]. The favorable effect of *Origanum majorana* on blood hematology also included raising the values of Hb, HCT, MCV, and lymphocyte percentage. The positive impact of OM on Hb may be related to its higher content of iron, which is considered an essential nutrient for hemoglobin production [47]. At the same time, the increased MCV suggests that OM has a hematopoietic impact, as RBCs (both new and young) are more prominent and contain a higher Hb amount [48]. According to Altan et al. [49], the H/L ratio is a valuable tool for explaining the different stress factors to which birds are exposed. In our experiment, adding OM powder to the pigeon diet decreased heterophils, increased lymphocytes, and, consequently, decreased the H/L ratio, which means that it may play a vital role in alleviating bird stress. Stef et al. [50] indicated that lymphocytosis could enhance interferon production. Furthermore, it was reported that hematological disorders caused by toxins, metals, or bacterial infections in different animals could be corrected by adding herbs to their diets [51,52].

### 4.5. Effect of Origanum majorana Powder on Biochemical Parameters, Inflammatory and Oxidative Markers

Total protein, albumin, creatinine, and urea were not changed by adding OM powder. Similarly, Shawky et al. [20] reported no significant difference in total protein and urea between the OM-supplemented group and the control group. Globulin was increased, while the ratio of albumin/globulin was decreased with OM powder. The same author suggested that the significant elevation in globulin indicates the *Origanum*’s ability to enhance the immunity of broiler chicks. *Origanum majorana* has been reported to induce hypocholesterolemia [15,31]. Our cholesterol result agreed with these reports. It was indicated that carvacrol and thymol present in OM could reduce cholesterol levels by inhibiting hepatic 3-hydroxy-3-methyl-glutaril198 CoA reductase [46,53].

In our experiment, interleukin 6 was not affected, while tumor necrosis factor-α levels tended to be increased with OM powder. Substances that promote leukocytosis may stimulate cytokine secretion from these cells, such as interleukin 6 and TNF-α [54]. Therefore, the tendency of TNFα to increase may be a compensatory reaction due to leukocytosis. Contrary to our results, Arranz et al. [55] indicated that the essential oil extracted from *Origanum majorana* has anti-inflammatory activity, as it contains terpineol and sabinene hydrate, which adversely affect cytokine production.

Malondialdehyde is considered a lipid peroxide. When its level increases, it can impair nucleic acid metabolism and function, destroy membrane proteins, and lead to autoimmune diseases [56]. To overcome lipid peroxidation and toxic free radicals, the secretion of some enzymes such as superoxide dismutase and glutathione peroxidase is enhanced, which play essential roles in the body’s defense mechanism against peroxidation. In the current study, OM powder significantly decreased the MDA level and increased the glutathione peroxidase level. The association between OM’s richness in phenolics and flavonoids (such as carnosol, carnosic acid, and hydroxycinnamic acid) and its antioxidant effect was investigated [9]. Therefore, *Origanum majorana* can play an essential role in maintaining the normal physiology, production, health, and welfare of animals.

### 4.6. Effect of Origanum majorana Powder on Duodenal, Cecal, and Bursal Follicle Histomorphometry

Feed utilization efficiency depends on feed digestion and absorption, which are affected by the intestinal surface [57]. Our histological results indicated that the intestinal villi length was slightly increased with OM powder. Abdelatty et al. [58] reported that improvements in growth performance are associated with increases in intestinal villi length and intestinal absorption. It was observed that the slight increase in intestinal villi length in the *Origanum* groups was associated with a numerical increase in body weight.

In the muscular layer, the cecal wall was thickened with OM powder supplementation. Scarce studies take morphometrical measurements for the cecum, despite its role in immune response, water absorption, digestion, and fermentation [59,60,61].

As a primary lymphoid organ, the bursa has a crucial role in B cell maintenance and establishment [41]. The area and number of follicles in the bursa of Fabricius were increased with OM powder. Attia et al. [62] reported that induction of humoral immunity and B lymphocyte production is associated with increases in the area of the bursal follicle.

### 4.7. Limitations of the Study

There was no possibility to confirm the results through q-PCR or to perform LC-MS analysis of serum, in order to see whether feeding OM increased the metabolites related to brain modulators, antioxidants, and immune modulators. In addition, the fear response was not assessed in our experiment in order to confirm as to whether it is correlated with increased feed intake. However, this research is essential and acts as the first step to realizing the influence of OM on improving the health status and welfare of pigeons; therefore, further experiments are required to emphasize the neural activity of birds.

## 5. Conclusions

Adding OM powder to the pigeon diet increased the relative weights of the lymphoid organs (spleen, thymus, and the number and area of bursal follicles), the WBC count, the lymphocyte count, and the serum globulin level. These effects suggest that OM powder may enhance bird immunity. The increased Hb, HCT, and MCV may have a hematopoietic effect. A decreased H/L ratio and MDA, as well as increased GPx, indicated that OM powder might have an antioxidant effect. Histological examination of the intestine suggested that nutrient absorption may be affected by adding OM powder, but this point needs further investigation. In conclusion, adding OM powder to the pigeon diet may play an essential role in alleviating stress, correcting some hematological disorders, and maintaining the physiology, metabolism, health, and welfare of birds, but more future work is still required.

## Figures and Tables

**Figure 1 life-13-00664-f001:**
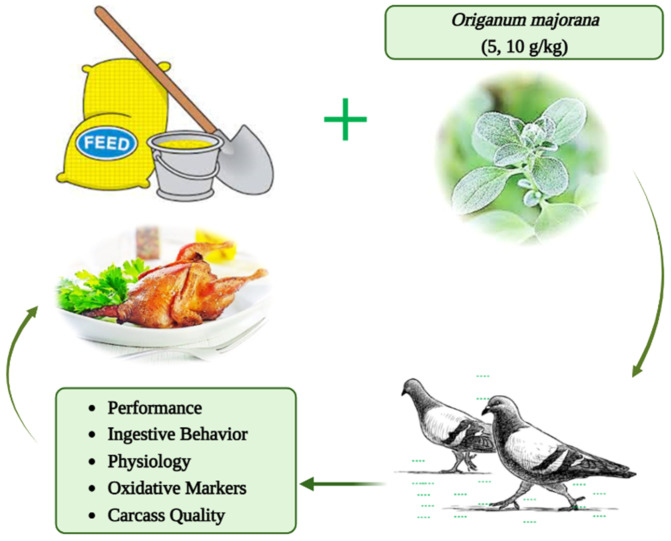
The schematic cartoon of the experimental design, created with BioRender.com.

**Figure 2 life-13-00664-f002:**
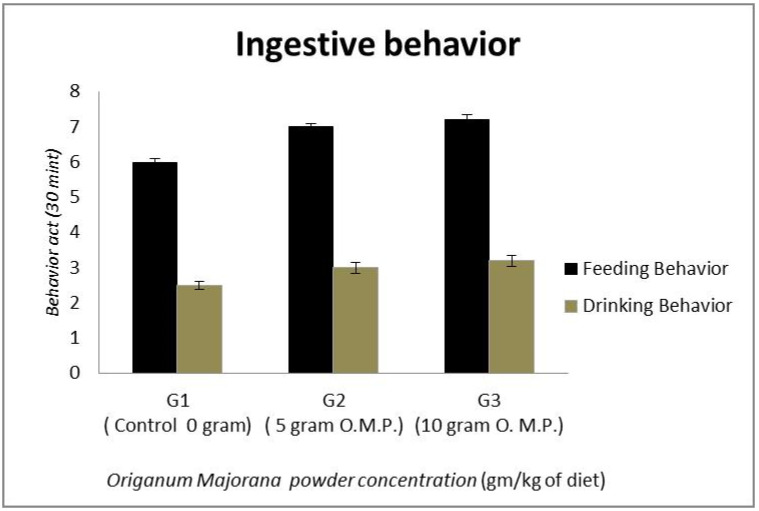
Effect of *Origanum majorana* powder on feeding and drinking behavior of pigeon squabs.

**Figure 3 life-13-00664-f003:**
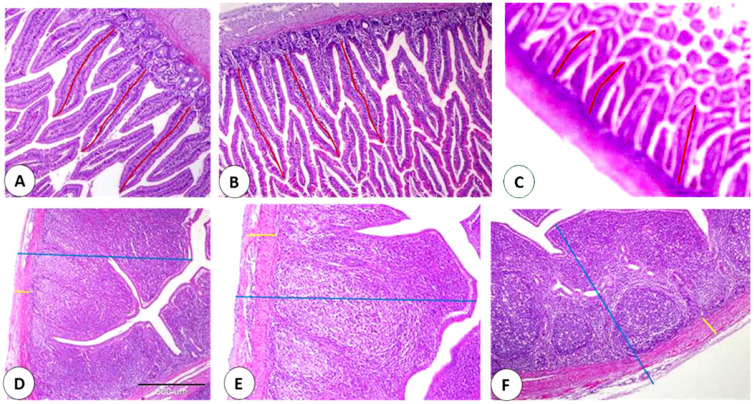
Effect of *Origanum majoran* powder on intestinal and cecal histology. Histomorphometry analyses of villi length of the duodenum [red line] in the control group (**A**), 0.5% *Origanum majorana* [OM] powder (**B**), and 1% OM powder (**C**). Histomorphometry analyses of the thickness of the cecal wall [blue line] and cecal muscular thickness [yellow line] in the control group (**D**), 0.5% OM powder (**E**), and 1% OM powder (**F**).

**Figure 4 life-13-00664-f004:**
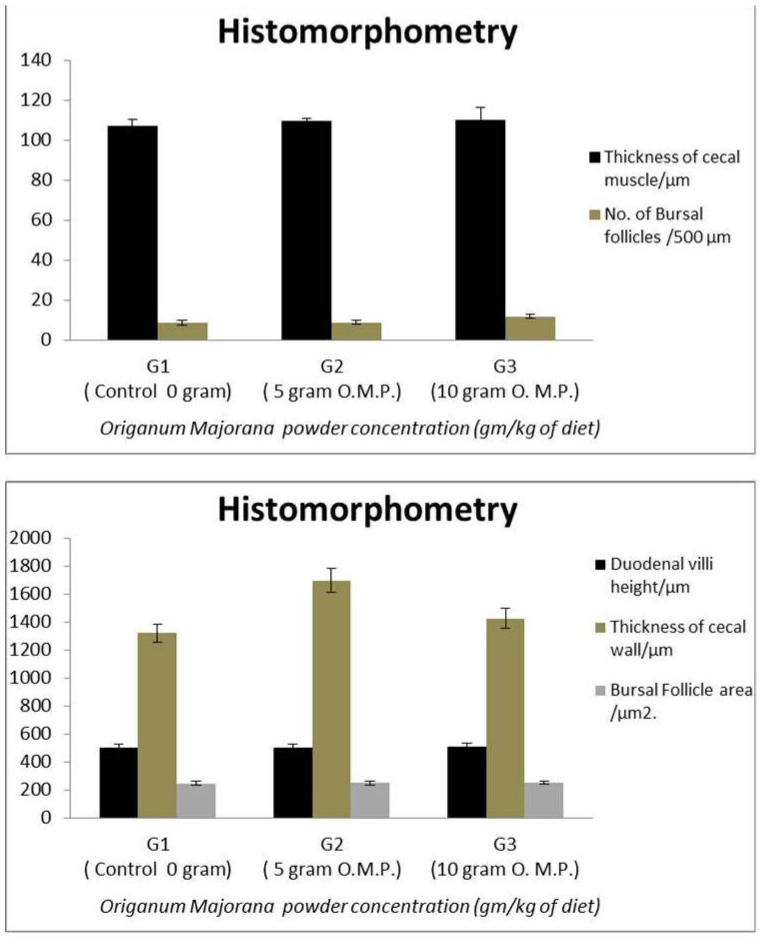
Histomorphometry using image j software showing; Thickness of cecal muscle/μm, No. of Bursal follicles/500 μm, Duodenal villi height/μm, Thickness of cecal wall/μm, Bursal Follicle area/μm^2^.

**Figure 5 life-13-00664-f005:**
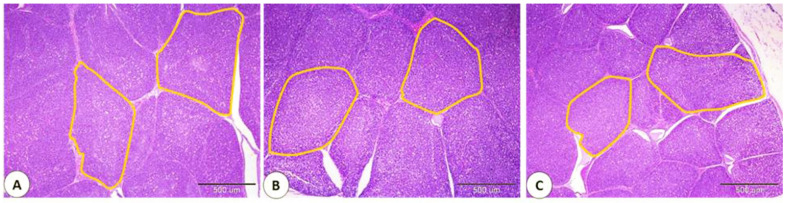
Effect of *Origanum majorana* powder on the bursa of Fabricius histology. Histomorphometry analyses of follicles area and the number in the control group (**A**), 0.5% *Origanum majorana* [OM] powder (**B**), and 1% OM powder (**C**).

**Table 1 life-13-00664-t001:** Ingredients and chemical composition of the grower basal diet of pigeon squabs.

Item	Control Diet *
**Ingredient, g/kg DM**	
Yellow corn ^1^	759.4
Soybean meal ^2^	197.2
Supplement ^3^	43.4
**Chemical composition, g/kg DM**	
CP	160
CF	28.8
EE	29.7
Available Ph	4
Ca	12
Lysine	8
Methionine	3
ME, Kcal/Kg diet ^4^	2988

* Control diet (grower basal diet) was fed to the three groups of pigeons by adding *Origanum majorana* powder at the level of 5 g/kg diet and 10 g/kg diet to the 2nd and 3rd groups of pigeons. ^1^ Composition (as fed basis): 89.41% DM, 9.50% CP, 2.11% CF, 3.70% EE and 3350 kcal/kg diet ME. ^2^ Composition (as fed basis): 87.97% DM, 44% CP, 6.50% CF, 0.8% EE, and 2230 kcal/kg diet ME. ^3^ Supplement consisted of (DM) 1.55% Dicalcium phosphate, 2.08% Ground limestone, 0.04% Methionine, 0.07% Lysine, 0.3% Salt, 0.3% Premix (supplied per Kg of dietary DM: Vit. A, 6,250,000 IU; Vit.D_3_, 2,500,000 IU; Vit. E, 25,000 mg; Vit.k_3_, 1750 mg; Vit.B_1_, 500 mg; Vit.B_2_, 2750 mg; Vit.B_6_, 1250 mg; Vit. B_12_, 10 mg; Nicotinic acid 20,000 mg; calcium pantothenate, 500 mg; Folic acid 500 mg; Biotin 50 mg; Iron 22 g; Copper 2.5 g; Zinc 37.5 g; Manganese 31 g; Iodine 650 mg; Selenium 113 mg; cobalt 50 mg). ^4^ Metabolizable energy was calculated using the ME of the ingredients, according to the previous report of NRC [18]. DM, Dry matter; CP, Crude protein; CF, Crude fiber; EE, Ether extract; Ph, Phosphorus; Ca, Calcium; ME, Metabolizable energy.

**Table 2 life-13-00664-t002:** Effect of adding *Origanum majorana* powder to pigeon diets on growth performance parameters.

Item	Treatment *			SEM	*p*-Value
	Control	0.5 M	1 M		
Initial BW, g	322	321	324	10.3	0.98
Final BW, g	424	438	439	13.8	0.68
BWG, g	102	115	117	10.5	0.57
Total FI, g	1178	1207	1237	41	0.62
FCR, g/g	4.7	5.8	5.2	0.87	0.73
EPI	12.8	12.5	12.2	1.97	0.98
RGR	17.5	19.2	19.1	2.34	0.85

* Treatment: Pigeons fed the Control diet (grower basal diet), pigeons fed the 0.5 M diet (grower basal diet with *Origanum majorana* powder at the level of 0.5%), and pigeons fed the 1 M diet (grower basal diet with *Origanum majorana* powder at the level of 1%). SEM, pooled Standard errors of means; BW, Body weight; BWG, Body weight gain; FI, Feed intake; FCR, Feed conversion ratio, EPI; European production index, RGR; Relative growth rate.

**Table 3 life-13-00664-t003:** Effect of adding *Origanum majorana* powder to the pigeon diet on carcass characteristics and immunity organs.

Item	Treatment *			SEM	*p*-Value
	Control	0.5 M	1 M		
Dressing %	71.38	69.34	69.14	0.79	0.17
**The relative weight of different organs [%]**					
Gizzard	1.86	1.82	1.88	0.04	0.54
Heart	1.37	1.29	1.29	0.07	0.6
Liver	1.71	1.65	1.52	0.08	0.32
Spleen	0.18 ^c^	0.26 ^b^	0.36 ^a^	0.02	0.01
Bursa of Fabricius	0.35	0.36	0.34	0.02	0.75
Thymus	0.49 ^b^	0.68 ^a^	0.68 ^a^	0.03	0.01

Means within the same row with different superscripts differ significantly (*p* < 0.05). * Treatment: Pigeons fed a control diet (grower basal diet), pigeons fed a 0.5 M diet (grower basal diet with *Origanum majorana* powder at the level of 0.5%), and pigeons fed 1 M diet (grower basal diet with *Origanum majorana* powder at the level of 1%).

**Table 4 life-13-00664-t004:** Effect of adding *Origanum majorana* powder to the pigeon diet on blood hematology.

Item	Treatment *			SEM	*p*-Value
	Control	0.5 M	1 M		
RBCs [×10^6^/mm^3^]	4.27 ^b^	4.43 ^b^	5.37 ^a^	0.145	<0.01
Hb [g/dL]	11.70 ^c^	12.30 ^b^	14.83 ^a^	0.135	<0.01
HCT [%]	38.90 ^c^	40.00 ^b^	48.70 ^a^	0.149	<0.01
MCV [fl]	89.15 ^c^	92.10 ^b^	96.55 ^a^	0.648	<0.01
MCH [pg]	27.63	27.76	27.96	0.652	0.93
WBCs [×10^3^/mm^3^]	38.33 ^b^	43.33 ^a^	44.00 ^a^	0.981	0.01
Monocyte %	7	7	8	0.577	0.42
Heterophil %	42.33 ^a^	27.33 ^b^	27.33 ^b^	2.769	0.01
Lymphocyte %	45.00 ^c^	73.00 ^a^	65.66 ^b^	1.981	<0.01
H/L ratio	0.95 ^a^	0.385 ^b^	0.42 ^b^	0.049	<0.01

Means within the same row with different superscripts differ significantly [*p* < 0.05]. * Treatment: Pigeons fed a Control diet [grower basal diet], Pigeons fed a 0.5 M diet [grower basal diet with Marjoram powder at the level of 0.5%], and Pigeons fed a 1 M diet [grower basal diet with Marjoram powder at the level of 1%].

**Table 5 life-13-00664-t005:** Effect of adding *Origanum majorana* powder to pigeon diet on blood biochemical parameters, and serum inflammatory and oxidative markers.

Item	Treatment *			SEM	*p*-Value
	Control	0.5 M	1 M		
Total protein, g/dL	5.05	5.37	5.3	0.16	0.37
Albumin, g/dL	3.4	3.23	3.25	0.08	0.34
Globulin, g/dL	1.65 ^b^	2.13 ^a^	2.05 ^a^	0.11	0.04
A/G ratio	2.06 ^a^	1.53 ^b^	1.59 ^b^	0.08	0.01
Cholesterol, mg/dL	194 ^a^	159 ^b^	148 ^b^	9.46	0.03
Urea, mg/dL	37.1	34.1	37.7	2.07	0.47
Creatinine, mg/dL	0.39	0.48	0.43	0.02	0.18
Interleukin 6 ng/L	273	284	279	23.2	0.95
TNFα Pg/mL	222 ^b^	230 ^a^	236 ^a^	6.9	0.09
MDA nmol/mL	8.95 ^a^	6.17 ^b^	4.50 ^c^	0.32	<0.01
GPx mu/mL	38.5 ^c^	102 ^b^	124 ^a^	5.57	<0.01

Means within the same row with different superscripts differ significantly [*p* < 0.05]. * Treatment: Pigeons fed a Control diet [grower basal diet], Pigeons fed a 0.5 M diet [grower basal diet with *Origanum majorana* powder at the level of 0.5%], and Pigeons fed a 1 M diet [grower basal diet with *Origanum majorana* powder at the level of 1%]. A/G ratio, Albumin/Globulin ratio; TNFα, Tumor Necrosis Factor α; MDA, Malondialdehyde; GPx, Glutathione Peroxidase.

## Data Availability

The datasets of the study are available from the corresponding author upon reasonable request.

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
