# Peer review of "Modulation of Immunity, Antioxidant Status, Performance, Blood Hematology, and Intestinal Histomorphometry in Response to Dietary Inclusion of Origanum majorana in Domestic Pigeons’ Diet"

_life, 2023, doi:10.3390/life13030664_

Round 1

Reviewer 1 Report

Hala et al., added OM powder to the pigeon diet improving their immunity, increasing their antioxidant status, and modulating some hematological disorders. The OM power is a promising additive for pigeons. However, there are several minor issues should be concerned as follow:  1. Please shorten the Title. 2.  In Figure 1, you should delete the last arrow. 3. Please re-edit the Figure 2, 3, 4. 4. Integrate the Figure 5-9 into one Figure. 5. In each table, IM should be 1M. 6. um2 should be μm2; also um 7.Please edit the paper by native English speeker.  

Author Response

The authors' response is uploaded. Please, check it and I hope to get your satisfaction.

Best regards,

Reviewer 2 Report

Dear Editor,

The manuscript is focused on pigeon health improvement through feeding medicinal plant. The author presented with lots of experimental work. It is interesting topic. However, I found some lacunae which leads to difficult to interpret the results. 

The diet preparation is not clear: how much gram of yellow corn, soybean meal and supplement were added per replicate to feed the pigeon. If you mentioned the quantity>>>>specify weight (in gram) or percentage (what basis it was calculated).

We can see that the improver text alignment found e.g., un-necessary space at line number 186 and other places. 

Adding OM to the bird’s diet alter the brain activity of poultry and reduces the stress which leads to increased feeding.

Since author says OM powder used in our experiment was too low to significantly affect pigeon performance or behavior, then then how it increases the organ weight of thymus and spleen. it is really contracting statement from author:>>>> they say that the thymus and spleen showed significant increases in their weights with the OM powder supply, which means that the pigeon immune responses were improved under the effect of Origanum majorana.>>>>>yes we agree, the OM induce the immune response but how it will increase the organ weight as reported by Ali [19] indicated that the spleen weight in broilers was not affected by adding variant Origanum majorana powder levels.

In addition, the author says at line 340>>> the total protein, albumin, creatinine, and urea were not changed by adding OM powder, >>>>>yes this is the evidence correlates that there is no weight increase in whole body or organ wise. Generally, protein and lipid build up increase the body weight but this study results are opposite. It is very difficult to agree the author statement.

Moreover, the author results mostly contrary to the previous report.

Only interesting information from author is that they reported logic behind biomarkers level, but author need to confirm the same through q-PCR.

I suggest the author to perform LC-MS analysis of serum to see whether feeding OM increased the metabolites related to brain modulator, antioxidant and immune modulators.

Author Response

The authors' response is uploaded. Please, check it and we hope we get your satisfaction!

Round 2

Reviewer 2 Report

Dear Editor,

The author trying to convence by their explanation but still very strongh evidence is required. As per the present data except some flaws as stated in the first revision, the MS can be accepted for publication. However, I advice the author to include their comments in the main manuscript that future work can be done for comments 3 and 6.

Author Response

Response Letter to Reviewer-2

A list of details of corrections and changes

 “Modulation of Immunity, Antioxidant Status, Performance, Blood Hematology, and Intestinal Histomorphometry in Response to Dietary Inclusion of Origanum majorana in Domestic Pigeons Diets”

We thank the respected editor and reviewers for their valuable comments to improve our study. All are well taken.

Reviewer-2:

General Comment

The manuscript is focused on pigeon health improvement through feeding medicinal plant. The author presented with lots of experimental work. It is interesting topic. However, I found some lacunae which lead to difficult to interpret the results.

Answer: Thanks for your valuable report; it helped us to improve our manuscript.

Specific Comments

1.      The diet preparation is not clear: how much gram of yellow corn, soybean meal and supplement were added per replicate to feed the pigeon. If you mentioned the quantity>>>>specify weight (in gram) or percentage (what basis it was calculated).

Answer: Thanks for your valuable note. As mentioned in the materials and methods section, birds were divided into 3 groups (the control, 0.5% OM, and 1% OM) Each group consisted of 18 birds. Then each group was divided into 3 subgroups (6/ each). The aim of subgrouping was to measure variability in the experiment and to make it easy to apply statistical tests. Each group was fed about 759.4 g yellow corn/kg DM, 197.2 g soybean meal/kg DM, and 43.4 g supplement/kg DM. The amount of feed was equally divided among the 3 subgroups in each group, which means that the amount of feed offered / replicate or subgroup was about 253.13 g yellow corn/kg DM, 65.73 g soybean meal/kg DM, and 14.47 g supplement/kg DM.

- The ingredients and chemical composition of the diet were changed from percent to g/kg DM in Table 1.  

(Line# 121).

2.      We can see that the improver text alignment found e.g., un-necessary space at line number 186 and other places.

Answer: Typing errors and un-necessary spaces were corrected using Grammarly software.

3.      Adding OM to the bird’s diet alter the brain activity of poultry and reduces the stress which leads to increased feeding.

Answer: Thanks for your comments. Yes, previous experiments indicated that aromatic plants like OM have a depressing effect on the neural activity through the activation of GABA receptors, which may lead to reducing the fear behavior with increasing the feeding times. Unfortunately, fear response was not assessed in our experiment to be correlated  with increased feed intake.

- Limitations of the study is written at the end of the manuscript. (Line# 376-383)

4.      Since author says OM powder used in our experiment was too low to significantly affect pigeon performance or behavior, then then how it increases the organ weight of thymus and spleen. it is really contracting statement from author: >>>> they say that the thymus and spleen showed significant increases in their weights with the OM powder supply, which means that the pigeon immune responses were improved under the effect of Origanum majorana. >>>>> yes we agree, the OM induce the immune response but how it will increase the organ weight as reported by Ali [19] indicated that the spleen weight in broilers was not affected by adding variant Origanum majorana powder levels.

Answer: Thanks for your comment - This sentence “OM powder used in our experiment was too low to significantly affect pigeon performance or behavior” was removed.

- The mechanism of action of OM powder in increasing the weight of thymus and spleen was not clear and it needs more investigation using more advanced techniques.  

- Wang et al., (2020) found that leukocytosis and suppressing oxidative stress caused modulation in spleen and thymus function in rats.

Ref#" Wang, Z., Lin, Y., Jin, S., Wei, T., Zheng, Z., & Chen, W. (2020). Bone marrow mesenchymal stem cells improve thymus and spleen function of aging rats through affecting P21/PCNA and suppressing oxidative stress. Aging (Albany NY), 12(12), 11386.

- In our study, by adding OM to pigeon diets, leukocytosis, decreasing the MDA level and increasing the glutathione peroxidase level were observed. Therefore, the effect of OM on thymus and spleen may be related to its effect on WBCs and oxidative stress.

- The variations between the results of our study and Ali (19) may be related to the variations in bird species, level of OM used, or diet composition.

5.      In addition, the author says at line 340 >>>  the total protein, albumin, creatinine, and urea were not changed by adding OM powder,  >>>>> yes this is the evidence correlates that there is no weight increase in whole body or organ wise. Generally, protein and lipid build up increase the body weight but this study results are opposite. It is very difficult to agree the author statement. Moreover, the author results mostly contrary to the previous report.

Answer:  Thanks for your valuable comments. As you mentioned the results of total protein, albumin, creatinine, and urea were correlated to the performance results. Regarding the weight of internal organs, it supported the performance, behavioral, and biochemical parameters results. The exception was in the weight of spleen and thymus and we discussed that in the previous point and supposed to be related to the effect of OM on leukocyte and oxidative stress.

- Regarding the previous reports, the variation may be mostly related to the difference in type of birds used as most of the previous studies used the broilers; however, pigeons were rare to be used. 

6.      Only interesting information from author is that they reported logic behind biomarkers level, but author need to confirm the same through q-PCR. I suggest the author to perform LC-MS analysis of serum to see whether feeding OM increased the metabolites related to brain modulator, antioxidant and immune modulators.

Answer:  Thanks for your notice. Unfortunately, we do not have enough samples to confirm our results using PCR or LC-MS. But in our future research plan, we will perform more experiments on OM powder using advanced techniques.

- Limitations of the study is written at the end of the manuscript. (Line# 376-383)

7.      NEW COMMENT

The author trying to convene by their explanation but still very strong evidence is required. As per the present data except some flaws as stated in the first revision, the MS can be accepted for publication. However, I advise the author to include their comments in the main manuscript that future work can be done for comments 3 and 6.

Answer:  Thanks for your comment. As per the reviewer advice, the limitations of the study is written at the end of the manuscript. (Line# 376-383).

Note-1: Please, we would like to add the word “diet”, to the same title as the one written for Ethical approval. So, We want to change the title from "Modulation of Immunity, Antioxidant Status, Performance, Blood Hematology, and Intestinal Histomorphometry in Response to Dietary Inclusion of Origanum Majorana in Domestic Pigeons diets" to "Modulation of Immunity, Antioxidant Status, Performance, Blood Hematology, and Intestinal Histomorphometry in Response to Dietary Inclusion of Origanum Majorana in Domestic Pigeons Diets".

Note-1: We acknowledge Dr. Mercia Nitzsche, Carshalton, United Kingdom‎‏, for the language editing. (line# 399, 400).

Thanks so much for your valuable notes to improve our manuscript.

Best regards,

Corresponding Author,

Prof. Dr. Ibrahim F. Rehan

Affiliation: *Department of Husbandry and Development of Animal Wealth, Faculty of Veterinary Medicine, Menoufia University, Shebin Alkom, Menoufia, 32511, Egypt. *Department of Pathobiochemistry, Faculty of Pharmacy, Meijo University, Yagotoyama 150, Tempaku-Ku, Nagoya-Shi, Aichi, 468-8503, Japan.

E-mail: ibrahim.rehan@vet.menofia.edu.eg

Tel.: +2-0100-2768-004
